



# Control of increased sedimentation on orogenic fold-and-thrust belt structure – Insights into the evolution of the Western Alps

Zoltán Erdős[1,2,3], Ritske S. Huismans[1], Peter van der Beek[2]

[1]Department of Earth Sciences, University of Bergen, Bergen, Norway

5    [2]ISTerre, Université Grenoble Alpes, Grenoble, France

[3]Now at Department of Geophysics and Space Science, Eötvös Loránd University, Budapest, Hungary

Correspondence to Zoltán Erdős (zoltan-erdos@caesar.elte.hu)



**Abstract** We use two-dimensional thermo-mechanical models to investigate the potential role of rapid filling of foreland basins in the development of orogenic foreland fold-and-thrust belts. We focus on the extensively studied example of the Western European Alps, where a sudden increase in foreland sedimentation rate is well documented during the mid-Oligocene. Our model results indicate that such an increase in sedimentation rate will temporarily

5    disrupt the formation of an otherwise regular, outward-propagating basement thrust-sheet sequence. The basement thrust active at the time of a sudden increase in sedimentation rate remains active for a longer time and accommodates more shortening than the previous thrusts. As the propagation of deformation into the foreland fold-and-thrust belt is strongly connected to basement deformation, this transient phase appears as a period of slow migration of the distal edge of foreland deformation. The predicted pattern of foreland-basin and thrust-front

10   propagation is strikingly similar to that observed in the North Alpine Foreland Basin and provides an explanation for the coeval mid-Oligocene filling of the Swiss Molasse Basin, due to increased sediment input from the Alpine orogen, and a marked decrease in thrust-front propagation rate. We also compare our results to predictions from critical-taper theory and we conclude, that they are broadly consistent, although, when sedimentation is included, critical-taper theory cannot be used to predict the timing and location of the formation of new basement thrusts.

15   The evolution scenario explored here is common in orogenic foreland basins; hence our results have broad implications for orogenic belts other than the Western Alps.



## 1. Introduction

The effects of surface processes on orogenic evolution have been intensively studied over the last three decades (e.g. Whipple, 2009). Numerous studies have shown that erosion can strongly influence the growth of orogenic hinterland regions with high erosion rates localizing deformation and creating a lower, narrower orogenic wedge (Beaumont et al., 1992; Braun and Yamato, 2010; Konstantinovskaia and Malavieille, 2005; Koons, 1990; Stolar et al., 2006; Willett, 1999).

Both numerical and analogue models also point towards a strong control exerted by synorogenic deposition on the structural development of orogenic forelands, as sedimentation rates affect the length of both thin- and thick-skinned foreland thrust sheets, as well as the amount of displacement taken up by individual faults (Adam et al., 2004; Bonnet et al., 2007; Duerto and McClay, 2009; Erdős et al., 2015; Fillon et al., 2012; Malavieille, 2010; Mugnier et al., 1997; Simpson, 2006a, b; Stockmal et al., 2007; Storti and McClay, 1995). However, direct comparison of model predictions with observations from natural case studies (e.g. Fillon et al., 2013) remains scarce.

The North Alpine Foreland Basin of France and Switzerland developed in response to continental collision in the Alps during early Tertiary time (Dewey et al., 1973; Homewood et al., 1986; Pfiffner, 1986). The stratigraphic infill of this foreland basin has been well documented (e.g. Sinclair, 1997; Berger et al., 2005; Kuhlemann and Kempf, 2002; Willett and Schlunegger, 2010 and references therein) and consists of two major stages: a Paleocene to mid-Oligocene deep marine (flysch) stage and a mid-Oligocene to late Miocene shallow marine and continental (molasse) stage (Fig. 1).

During the first stage, exhumation rates of the orogenic hinterland and deposition rates in the foreland basin were low; hence the basin remained underfilled (Allen et al., 1991; Burkhard and Sommaruga, 1998; Sinclair and Allen, 1992). At the onset of the second stage, both erosion rates in the Alps and deposition rates in the foreland basin increased (Schlunegger et al., 1997; Schlunegger and Norton, 2015; Sinclair and Allen, 1992), creating a filled to overfilled foreland basin. The transition from an underfilled to an overfilled state coincided with a marked decrease in thrust-front advance rate (Sinclair and Allen, 1992), but the links between the two have remained speculative.

Here, we use numerical models that build on our previous work (Erdős et al., 2015; Erdős et al., 2014), to test how an increase in sedimentation rate affects mountain-belt and foreland fold-and-thrust belt evolution. In earlier work (Erdős et al., 2015) we showed how our model predictions were consistent with minimum-work theory. Here, we quantitatively compare our models to critical-taper theory in order to assess the predictions of this simple but widely used model, when including a more complex and realistic rheology. Our main aim is to explore the potential causal relationship between a sudden increase in sediment influx and the temporary slowing of thrust-front propagation, as observed in the North Alpine Foreland Basin. Such a sediment accumulation scenario is common in foreland basins (e.g. Allen and Homewood, 1986), hence the demonstration of a causal relationship should have significant impact on our understanding of not just the North Alpine Foreland, but the development of similar orogenic systems around the world.



## 2.  Numerical method

We explore the potential links between syntectonic sedimentation and orogen structure through the use of 2D arbitrary Lagrangian-Eulerian thermo-mechanical modeling (Erdős, 2014; Thieulot, 2011) coupled to a simple surface process algorithm. The numerical experimental setup is very similar to the one we used in our previous studies (Erdős et al., 2015; Erdős et al., 2014) and is explained in detail in the Supplementary material.

The thermo-mechanical model consists of strain weakening frictional-plastic materials that allow for localization of deformation (e.g. Huismans et al., 2005). Our experiments use a 4-layer crust-mantle rheology in which the upper and lower crust, as well as the upper lithospheric mantle undergo frictional plastic deformation, while the middle crust and lower lithospheric mantle exhibit power-law viscous creep (Fig. 2). A 3-km thick pre-orogenic sediment package at the top of the model is separated from the crust by a 1-km thick weak layer representing a décollement horizon (e.g., an evaporite or shale layer). In order to include self-consistent inherited extensional weakness zones, the model is first extended, before the velocity boundary conditions are inverted to create a contractional regime (e.g. Erdős, 2014; Jammes and Huismans, 2012).

The surface-process model includes an elevation-dependent erosion algorithm as well as a sedimentation rule that fills topography up to a reference base level at each time step. Both the erosion and sedimentation algorithms are simple and do not conserve mass; however, the resulting basin-fill geometries are consistent with observations from natural foreland-basin systems (DeCelles and Giles, 1996).

The model experiments presented here have sufficiently high resolution (500 m horizontally and 200 m vertically in the upper crustal domain) to bridge the large range of scales from the entire collisional orogen to the fold-and-thrust belt and the interaction with synorogenic deposition.

## 3.  Model results

We present three model experiments that demonstrate the response of crustal deformation to sudden temporal changes in synorogenic sedimentation. For Model 1, neither erosion nor sedimentation is included (Fig. 3a-d). In Model 2, a simple elevation-dependent erosion model is applied together with fixed base-level sedimentation (Erdős et al., 2015): during each model time-step, basins are filled with sediments to a prescribed base level (Fig. 3e-g). Model 3 is identical in setup to Model 2, but sedimentation is initiated 10 Myr earlier and the base level of sedimentation is increased during the experiment (15 Myr after initiation) to mimic the transition from an underfilled to an overfilled foreland basin (Fig. 3h-i; see also supplementary animations) as observed in the Western Alps (e.g. Sinclair and Allen, 1992).

### 3.1.  Model 1.

During the 15 Myr of initial extension, a broad, approximately 200-km wide asymmetric rift basin is formed in the center of the model domain, consisting of a number of rotated crustal blocks with mantle material reaching the surface at two different locations, approximately 50 km apart (Fig. 3a). This is followed by a 15 Myr long inversion period culminating in subduction initiation and the formation of an uplifted central block (key-stone structure) with a



distinct internal structure consisting of a number of inverted normal faults around a core of uplifted lower crustal and lithospheric mantle material (see Supplementary Movie 1).

In the third phase of Model 1, deformation migrates into the subducting plate, building up the pro-wedge initially by formation of a crustal-scale pop-up structure, and then primarily through an outward-propagating sequence of

basement thrust sheets (Fig. 3d) with an average thrust sheet length of 52 km. We use the term basement thrust sheet when referring to thrust sheets that cut the crystalline basement (upper crust). Superposed on this sequence, and often spatially slightly ahead of it, the pre-orogenic sediments are also deformed, creating a complex thin-skinned fold-and-thrust belt (Fig. 3b-c; for an extensive description of the interaction of thin-skinned and thick-skinned deformation see Erdős et al., 2015). Deformation in the retro-side of the orogen remains comparatively subdued

throughout the model but the initially uplifted central block, which includes a lower crustal/ lithospheric mantle core, is transported more than 50 km onto the overriding plate.

### 3.2.  Model 2, with erosion and sedimentation

The surface process algorithms in Models 2 and 3 are activated at 45 Myr and 35 Myr respectively. Consequently, all presented models exhibit the same behavior during the first two phases described above.

Following initiation of erosion and sedimentation at 45 Myr in Model 2, sediment-loaded foreland basins form on both sides of the orogen, with more intense thin-skinned deformation on the pro-side. The sequence of outward-propagating basement thrust sheets in the pro-wedge is disrupted as deformation remains localized on the active frontal basement thrust for 8 Myr, instead of the 4 Myr observed in Model 1, before stepping out below the foreland basin 13 km further than in Model 1 (Fig. 3e-g; Supplementary Movie 2).

The effect can be well illustrated by comparing the length and displacement of basement thrust sheets around the time of the onset of sedimentation (Fig. 4). Prior the onset of sedimentation, *Thrust A* accumulated 6 km displacement before *Thrust B* created a new, 45-km long basement thrust sheet (Fig. 4a). After the onset of sedimentation, *Thrust B* remained active for about 8 Myr and accumulated 24 km displacement before *Thrust C* created a new, 83-km long thrust sheet in the footwall of *Thrust B* (Fig. 4b).

As the model progresses further, upper crustal blocks in the internal parts of the orogen that were initially covered with pre-orogenic sediments are deeply eroded, reaching the surface and bringing the lower crustal/mantle lithospheric core of the central block to shallow depths. A small sliver of mantle lithospheric material eventually reaches the surface along a back-thrust (Fig. 3f).

We recorded maximum sedimentation rates for 2 Myr intervals (see the alternating orange and green layers of syn-

tectonic sediments on Fig. 3e-i) throughout the model. After an initial peak of 2.7 km $Myr^{-1}$ between 45 and 46 Ma, when the entire available accommodation space is filled up to the prescribed baselevel, the maximum sedimentation rates in the pro-foreland basin stabilized around an average of 0.45 km $Myr^{-1}$.



### 3.3. Model 3, with erosion and intensifying sedimentation

The evolution of Model 3 is very similar to that of Model 2, even though sedimentation and erosion start 10 Myr earlier. Significant differences can only be seen between the pro-foreland basins, after the base level of sedimentation is raised (simulated here by an increase in the sedimentation base level over a 0.5 Myr period) to

mimic the transition from an underfilled to an overfilled foreland basin (Fig. 3h-i, Supplementary Movie 3). The base-level change results in a temporary (approximately 2 Myr long) increase in the maximum sedimentation rate in the foreland basin (from an average of 0.45 km Myr$^{-1}$ to 1,1 km Myr$^{-1}$ at the location of the frontal thrust). Subsequently, the maximum sedimentation rate quickly decreases to its previous (average) value.

As observed in Model 2, the initiation of sedimentation alters the architecture of the orogenic foreland by creating

longer basement thrust sheets. Similarly, a sudden increase in the sedimentation rate in Model 3 also results in a change in the foreland development.

Again, this can be well illustrated by looking at the deformation pattern around the time of increase in sedimentation rate (Fig. 5). Prior the increase in sedimentation rate, *Thrust A* accumulated 10 km displacement before *Thrust B* created a new, 45-km long basement thrust sheet (Fig. 5a). After the increase in sedimentation rate, *Thrust B*

remained active for about 8 Myr and accumulated 22 km displacement before *Thrust C* created a new, 75-km long thrust sheet in the footwall of *Thrust B* (Fig. 5b).

The subsequent basement thrust-sheet sequence consists of longer thrust sheets (on average 45 km instead of the previous 40 km) that are active for longer times (on average 6.5 Myr instead of 4 Myr), compared with the model behavior before the increase in sedimentation rate (Fig. 3h-i; Supplementary Movie 3).

## 4. Discussion

The first-order evolution of all three presented models is similar, regardless of the imposed erosion/sedimentation scenario. First an asymmetric rift is formed, followed by the inversion of the large normal faults. After full inversion, a central key-stone structure is uplifted, with a crustal-scale thrust on either side of it. One of these basement thrusts then becomes the locus of subduction. After the polarity of subduction is established, a new basement thrust is

formed in the subducting pro-wedge lithosphere on average every 3.1 Myr (in case of Model 1) in an outward-propagating sequence. The main differences between the models are the position and timing of thrust activations.

The step-like migration of the deformation front is present throughout all our model experiments, but it is enhanced when a change in the sedimentation history occurs. In Model 2, the distal edge of the foreland basin advances rapidly after the onset of sedimentation, while the basement-deformation front remains stationary (Fig. 6b). After this

transitional period, lasting about 2 Myr, a new propagation order is established with longer basement thrust sheets (on average 46 km instead of 40 km) that stay active for longer times (on average 7 Myr instead of 4.5 Myr). In Model 3, two such transitional periods can be observed (Fig. 6c) one at the onset of sedimentation (35 Ma; see caption) and one at the increase in sedimentation rate (20 Ma; see caption). During this latter transition, the distal edge of the foreland basin rapidly advances again (approximately 150 km in 2.5 Myr), while the outermost basement

thrust remains active for 4 Myr longer than the previous frontal thrusts (7.5 Myr instead of the previous 3.5 Myr).



In general, the location of a newly initiated in-sequence basement thrust corresponds to the point where the total work needed to slide on the viscous mid-crustal weak zone and to break through the upper crust is lower than the work needed to maintain deformation on the existing thrust front (Erdős et al., 2015; Fillon et al., 2012; Hardy et al., 1998). Upon initiation (or increase) of sedimentation in the foreland basin, the work required to create a new

basement thrust is suddenly increased as the sediments effectively expand the thickness of the rock column overlying the mid-crustal weak zone (Erdős et al., 2015). This increased resistance against the formation of a new thrust breaks the previous cyclic behavior and delays the propagation of the deformation front into the foreland basin.

## 4.1. Comparison with the Alps

The models presented here capture a number of first-order features of the Western European Alps (Schmid and

Kissling, 2000; Schmid et al., 2017) (Fig. 1), including: (a) a major step in Moho depth between the European and Adriatic (or Apulian) plate; (b) strong decoupling between the upper and lower crust, with the lower crust underthrusting and subducting with the mantle lithosphere; (c) stacking of basement thrust sheets in the central part of the orogen; (d) shallow emplacement of lithospheric mantle material in the retro-wedge, with a sliver of mantle material reaching the surface (Fig. 3f, g, i), loosely resembling the Ivrea body and Sesia-Lanzo zone, respectively; and (e) a

generally asymmetric orogen with deformation stepping out much further on the pro-side than on the retro-side. The presence of a weak décollement below the pre-orogenic succession in this model is also characteristic of the Western Alpine foreland and allows for the coexistence of thin-skinned and thick-skinned tectonics (see e.g. Erdős et al., 2015), a feature that is much less prominent in the Eastern Alps, where the décollement is absent (Schmid et al., 2004).

The initial extensional phase allows the creation of physically self-consistent inherited structural weakness zones, as observed in most orogens. After extending the model for 15 Myr, the continental lithosphere has effectively ruptured, creating two small separate ocean basins that mimic the pre-orogenic presence of the Piemont-Ligurian and Valais basins in the Alpine domain (Stampfli et al., 2001). It must be pointed out that running the models further in extensional mode in this setup is not viable because there is no built-in mechanism for the creation of oceanic

lithosphere. The effects of a thermal relaxation phase were not explored either, as potentially important mechanisms like strain-healing are not yet implemented in the model.

The basement under the pro-foreland basin is rather smooth, dipping on average 3° towards the orogen at the time-slice captured on figure 4b (which corresponds best to the present state of the North Alpine Foreland Basin). This value is in good agreement with those inferred from the interpretation of seismic reflection lines (Burkhard and

Sommaruga, 1998; Sommaruga, 1999).

The increase of sedimentation in Models 2 and 3 links thrust-front propagation and the onlap of sediments onto the foreland as observed in the North Alpine Foreland Basin (e.g. Sinclair, 1997; Fig. 5). Both deposition scenarios lead to longer frontal basement thrusts that remain active for a longer period before a new thrust is formed. This suggests that increased sedimentation, which resulted from the increasing relief and changing climate in the Alpine hinterland

(Schlunegger et al., 1997; Schlunegger and Norton, 2015), was a significant factor in the mid-Oligocene stalling of thrust-front advance observed in the western section of the North Alpine Foreland. Note that this behavior is not




observed further east along the foreland where the amount of orogen perpendicular shortening is less and the decoupling salt-layer is absent from the foreland basin (Schmid et al., 2004). This could well limit the distance to which the thin-skinned deformation of the foreland fold-and-thrust belt can reach.

We also note that the stepwise behaviour shown by Sinclair (1997) is present in our models even if there is no
change in the deposition scenario applied. However, we argue that an increase in the amount of material deposited in the foreland basin will necessary result in stalling of the thrust-front propagation while it will also allow for the distal edge of the foreland basin to migrate further onto the down-going plate.

### 4.2. Comparison with critical-taper theory

We attempt to explain the observed behaviour of our models at the scale of the entire wedge in terms of critical-taper
theory (Chapple, 1978; Dahlen, 1990; Davis et al., 1983). According to this theory, a wedge will evolve towards a critical state, characterized by being at the verge of brittle failure both internally and at its base. As a consequence, equilibrium is reflected by a self-similarly growing wedge with a stable surface slope ($\alpha$) and detachment dip ($\beta$) (Davis et al., 1983); such a wedge should react instantaneously to changes in stress regime. Lateral variations in the structure and surface slope of European Alpine foreland have been explained using critical-taper theory (von Hagke
et al., 2014 and references therein). However, this purely brittle mechanical theory has limited applicability to our model due to the presence of viscous-plastic deformation and strain-weakening materials (Buiter, 2012; Simpson, 2011). Simpson (2011) argued that an elastic-plastic wedge is often well below the critical stress threshold locally. Hence, we explore here whether the large-scale deformation of our model orogens exhibit a behaviour that is consistent with critical-taper theory predictions.

When we consider a brittle Coulomb wedge, a sudden increase in sedimentation rate will result in the filling up of the previously unfilled (or underfilled) foreland basin, reducing $\alpha$ significantly while moderately increasing $\beta$ due to the loading of the basin. Critical-taper theory predicts that such a sudden change in the taper angles, without a simultaneous modification of the mechanical properties of the wedge or the basal detachment, should drive the wedge towards a sub-critical state. Subsequently, the wedge needs to deform (thicken) internally to increase its taper
angle until it reaches critical state once again (see also Willett and Schlunegger, 2010).

We analyze five models to assess whether our pro-wedges replicate the above predictions of critical-taper theory. In order to isolate the potentially tangled effects of erosion and sedimentation, we include in this analysis a model with erosion but no sedimentation (Model 1.1) and one with sedimentation but no erosion (Model 2.1). We define the wedge as the zone between the surface trace of the frontal (thin skinned) thrust and the lower-crustal indenter of the
overriding plate (denoted S-point on Fig. 7). The basal slope $\beta$ is calculated using the top of the lower crust as a reference horizon. We acknowledge that these definitions are arbitrary and in some cases at odds with assumptions of critical-taper theory (i.e. the top of the lower crust separates the ductile middle crust and the brittle lower crust) but these definitions allow for a consistent derivation of $\alpha$ and $\beta$ values for each time slice in every model.

Due to the complexity of the surface topography (and – to a lesser extent – the basal décollement), representing the
entire wedge with a single $\alpha$-$\beta$ pair is notoriously difficult. In this study, we calculated multiple sets of $\alpha$ and $\beta$ values along the wedge using a range of different sampling intervals for every time slice of the model (e.g., Fig. 7).



Subsequently we calculated the mean α and β values for each sampling interval and visualized the resulting mean of these sampling intervals using boxplots (see Fig. 8). This analysis allows us to identify temporal trends that are persistent through a range of characteristic length scales. We have tested over a hundred different sampling intervals from 2.5 km to 100 km and decided to use a subset of 41 of these, ranging from 10 to 30 km, to create the plots of

this study. Note that the trends described here were also present at the higher and lower ends of the sampling scale.

For brevity, we only discuss here the implications of the above detailed critical-taper analysis. The individual α, β and α + β vs. model-time plots and their detailed interpretations can be found in the Supplementary Material, along with a detailed description of models 1.1 and 2.1. Generally, the models without sedimentation conform to the predictions of critical-taper theory. After an initial mountain-building phase, α + β stabilizes at a roughly stable level

and is only slightly perturbed around individual basement thrusting events (see Fig. 8a). Erosion slightly increases α and reduces β, keeping α + β at a constant value. The increase in α is a result of the development of a narrower and steeper wedge with a narrower foreland basin. Conversely, the decrease in β is partly due to decreased topographic loading: models with erosion do not produce topography higher than 6 km, while models without erosion can grow topography as high as 8 km.

When sedimentation is included in the models, the behavior is considerably more complex with the importance of the initiation of new thin-skinned frontal thrusts becoming more pronounced (Fig. 8b and c). As the orogenic foreland – and hence the wedge itself – grows wider, the crustal load exerted by the orogen grows as well. The loading increases β until the deformation moves to a new frontal thrust, further widening the wedge and incorporating a previously undeformed, gently dipping basement, which instantaneously reduces β. These cycles in β

are superimposed on top of a long-term decreasing trend, likely resulting from the wedge becoming larger, warmer and easier to deform over time. In the meantime, the wide and low-relief orogenic foreland thrust belts generally decrease α to very low (0.5°-2°) values.

The observed cyclic behavior, in which the deformation periodically migrates to a new frontal thrust is similar to the "punctuated thrust deformation" described by Hoth et al. (2007) and Naylor and Sinclair (2007), whereby the

position of the deformation-front fluctuates as successive thrusts are gradually incorporated into the wedge. This discrete, punctuated behavior causes the wedge to oscillate around a critical taper value rather than stay in complete equilibrium through time. Here we have shown, moreover, how erosion and sedimentation influence this behavior consistent with the predictions of critical-taper theory.

We have created animations showing the temporal and spatial (along-profile) variations of α, β, an arbitrary shallow

metric of the strain rate and the topography for models 1 and 2 (see Supplementary Movies 6 to 9). Our aim with this exercise was to establish whether the changes in topography (α, β) are driven by strain rate changes or the other way around. A key observation here is that the α evolution of Model 1 (and to a lesser extent of Model 2) shows a particular pattern: a new thrust is activated after the α of the region around the active fault reaches ~10°. After the new thrust is activated, this high α rapidly decays. This suggests that α ≈ 10° can locally be seen as a critical value,

which triggers the formation of a new frontal thrust. This new thrust is generally activated close to the tip of the active thin-skinned deformation front.

When sedimentation is included (Model 2), the high-α regions are more persistent. We argue that, since the sediments are stifling the foreland basin, there is significantly less room for thin-skinned deformation that would





otherwise create a gentler slope around the surface trace of the basement thrusts. This results in negative-α basins sliding between thick-skinned thrusts on top of the décollement. Our thermomechanical models are therefore in agreement with the analytical results shown by Willett and Schlunegger (2010).

### 4.3. Implications for other mountain belts

An early synthetic stratigraphic model of foreland-basin development (Flemings and Jordan, 1989) showed that peripheral orogenic foreland basins have a tendency to evolve from an underfilled into a filled to overfilled state. Numerous studies focusing on the stratigraphic infill of natural foreland basins (e.g. Allen et al., 1991; DeCelles and Burden, 1992; Quinlan and Beaumont, 1984) have demonstrated the merits of this model. Moreover, as the internal part of the orogen grows, more surface area reaches higher elevations, resulting in a potential increase in erosion rates and consequently, sediment flux into the foreland basin (Simpson, 2006a, b; Sinclair et al., 2005). Hence the orogenic foreland-basin evolution scenario described in this study should be applicable to a wide range of orogens around the globe. A prime example may be the Southern Pyrenean (pro-) foreland fold-and-thrust belt, where a middle Eocene increase in sedimentation rate was accompanied by stalling of the thrust front (Sinclair et al., 2005). Based on their stratigraphic models, Flemings and Jordan (1989) proposed that changes in the rate of thrust loading, climate, or source-rock lithology (all present in their models through surface-process transport coefficients) can cause the shift from underfilled to overfilled basins. Our model results imply that there is a strong feedback between these potential controls and the state of the basin fill.

### 5. Conclusions

The thermo-mechanical models presented here provide first-order insights into the intricate relationship between changing sedimentation rates and deformation patterns in orogenic forelands. Our models show that a sudden increase in sedimentation rate disrupts thrust-front and foreland-basin propagation patterns. The outermost basement thrust remains active for a significantly longer time and accumulates more deformation than previous thrusts developed during periods of lower sediment input, before deformation steps out again under the sediment-loaded foreland basin. After determining α and β values for each model and examining their evolution over time, we conclude that they are broadly consistent with predictions from critical-taper theory, despite the more complex and realistic rheology included in our models. However, when sedimentation is included, critical-taper theory cannot be used to predict the timing and location of the formation of new basement thrusts.

The results are in good agreement with observations from the Western Alps and the North Alpine Foreland Basin, where deformation remained relatively stable for an extended period of time after the foreland basin shifted from an underfilled to a filled/overfilled state.



## 6. Author contribution

Zoltán Erdős and Ritske S. Huismans designed the experimental setup. Zoltán Erdős ran the model experiments and all three authors contributed in the interpretation of the results. Zoltán Erdős prepared the manuscript with contributions from all co-authors.

**Acknowledgements**

We thank Fritz Schlunegger for his constructive comments at an early stage of this project and Stefan Schmid for providing the latest version of the Alpine cross-section assembled by his group. We also thank Mary Ford and Christoph von Hagke for their constructive feedback on a previous version of this manuscript.

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





**Figure captions**

**Figure 1**: Western Alpine cross-section and geological map redrawn after Schmid and Kissling (2000) and Schmid et al. (2017) with the inset showing the section interpretation of Roure (2008) along part of the same deep-seismic section. SL indicates the location of the Sesia-Lanzo zone.

**Figure 2: (a)** Model geometry showing layer thicknesses (including a close-up of the crust), the position and size of the weak seed (pink square), the lateral velocity boundary conditions (black arrows along the sides of the box; note the ± sign), and the initial strength and temperature profiles of the models. The material properties corresponding to each layer (including the syn-tectonic sediments) are presented in Table 1. **(b)** Frictional-plastic strain softening is
achieved through a linear decrease of $\phi_{eff}$ from 15° to 2° with a simultaneous decrease of C from 20 MPa to 4 MPa. **(c)** Legend for materials shown in (a).

**Figure 3**: Model results. The material coloring scheme is identical to that used in Figure 1. All models are run for 65 Myr: 15 Myr (150 km) extension followed by 50 Myr (500 km) contraction for a total net contraction of 350 km. **(a)-**
**(d):** Model 1 with no surface processes, showing deformed Lagrangian mesh and isotherms after **(a)** 15 Myr ($\Delta x$ = -150 km) and **(d)** 65 Myr ($\Delta x$ = 350 km) respectively. **(b)** and **(c)** are extracts from panel **(d)** showing the small-scale deformation patterns in the foreland fold-and-thrust belts. **(e)-(g):** Model 2 including a simple surface-process algorithm filling up accommodation space until a baselevel of -500 m, showing deformed Lagrangian mesh and isotherms after 65 Myr ($\Delta x$ = 350 km). **(e)** and **(f)** are extracts from panel **(g)** showing the small-scale deformation
patterns in the foreland fold-and-thrust belts. **(h)-(i):** Model 3 including a simple surface-process algorithm with the sedimentation baselevel changing from -500 m to 0 m at t = 45 Myr Panels show deformed Lagrangian mesh and isotherms after 65 Myr ($\Delta x$ = 350 km). **(h)** is an extract from panel **(i)** showing the small-scale deformation patterns in the pro-wedge foreland fold-and-thrust belt.

**Figure 4:** The evolution of Model 2 around the time of the onset of sedimentation (and erosion). The material coloring scheme is identical to that used in Figure 1. **(a)** Model 2 at 45 Myr ($\Delta x$ = 150 km), just before the onset of sedimentation. White marks show the length of the active external basement thrust sheet (thrusting along *Thrust B*). The length is measured using the visugrid (black grid advected with the materials in the model); we counted the number of undeformed cells in the top row in the basement between the old and the new frontal thrust. Red marks
show the amount of displacement along the last abandoned thrust (*Thrust A*). **(b)** Model 2 at 53 Myr ($\Delta x$ = 230 km) at the time of the initiation of the first basement thrust-sheet after the onset of sedimentation. White marks show the length of the active external basement thrust sheet (thrusting along *Thrust C*). Red marks show the amount of displacement along the just abandoned thrust (*Thrust B* corresponding to *Thrust B* of Figure 3a). Further towards the orogenic hinterland the over-steepened *Thrust A* is shown (corresponding to *Thrust A* of Figure 3a).


**Figure 5:** The evolution of Model 3 around the time of increase in sedimentation rate. The material coloring scheme is identical to that used in Figure 1. **(a)** Model 3 at 51 Myr ($\Delta x$ = 210 km), at the time of increase in sedimentation

rate. White marks show the length of the active external basement thrust sheet (thrusting along *Thrust B*). The length calculation method is the same as in Figure 3. Red marks show the amount of displacement along the last abandoned thrust (*Thrust A*). **(b)** Model 3 at 59 Myr (Δx = 290 km) at the time of the initiation of the first basement thrust-sheet after the increase in sedimentation rate. White marks show the length of the active external basement thrust sheet

(thrusting along *Thrust C*). Red marks show the amount of displacement along the just abandoned thrust (*Thrust B* corresponding to *Thrust B* in Figure 3a). Further towards the orogenic hinterlands the gradually steepening *Thrust A* is shown (corresponding to *Thrust A* in Figure 3a).

**Figure 6**: Thrust-front propagation and sediment onlap on the distal edge of the foreland basin vs. time **(a)** in the

Western Alps (redrawn after Sinclair (1997); **(b)** derived from Model 2; and **(c)** derived from Model 3. The thin dashed line in **(b)** and **(c)** shows the thrust-front propagation pattern of Model 1. Note, that in **(b)** and **(c)** the time axis of the models are reversed from Myr (forward model time) to Ma (time before "present") to fit the original axis of the Western Alps.

**Figure 7**: Example of α and β sampling routine. *S-point*: internal limit of the wedge considered for critical taper

analysis, located at the tip of the lower-crustal indenter of the overriding plate. *Wedge-tip*: the outer tip of the wedge considered for critical taper analysis, located at the tip of the orogenic deformation zone. *Red dots*: elevation sampling points along the wedge for a given sampling interval. For each sampling interval, α is first calculated for every adjacent point (e.g. $α_{11}$, $α_{12}$) before we calculate the mean ($α_1$) of these local, individual α-values for the entire

wedge. The process is then repeated for all sampling intervals (e.g. $α_{21}$). *Blue dots*: depth sampling points along the wedge for a given sampling interval. β is calculated in the same manner as α (described above).

**Figure 8:** Plots of α + β vs. model time for models 1 **(a)**, 2 **(b)**, and 3 **(c)**. For each time-slice, the α and β values were determined using a range of sampling intervals. The boxplots present the average α + β, α and β values of these

individual sampling intervals calculated for the entire wedge. On each box, the central mark is the median, the edges of the box are the 25th and 75th percentiles, the whiskers extend to the most extreme data-points considered not to be outliers. The outliers are plotted individually.

**Table 1:** Mechanical and thermal parameters used in the models for each material



**Supplement 1 – Detailed numerical methods**

**Supplement 2 – Additional model descriptions**

**Supplement 3 – Description of individual α, β, and α + β plots**

**Supplement 4 – Model movies**

5   **Movie A1**: Evolution of Model 1

**Movie A2**: Evolution of Model 2

**Movie A3**: Evolution of Model 3

**Movie A4**: Evolution of Model 1.1

**Movie A5**: Evolution of Model 2.1

10   **Movie A6:** α, shallow strain rate and topographic evolution of Model 1

**Movie A7:** β, shallow strain rate and topographic evolution of Model 1

**Movie A8:** α, shallow strain rate and topographic evolution of Model 2

**Movie A9:** β, shallow strain rate and topographic evolution of Model 2





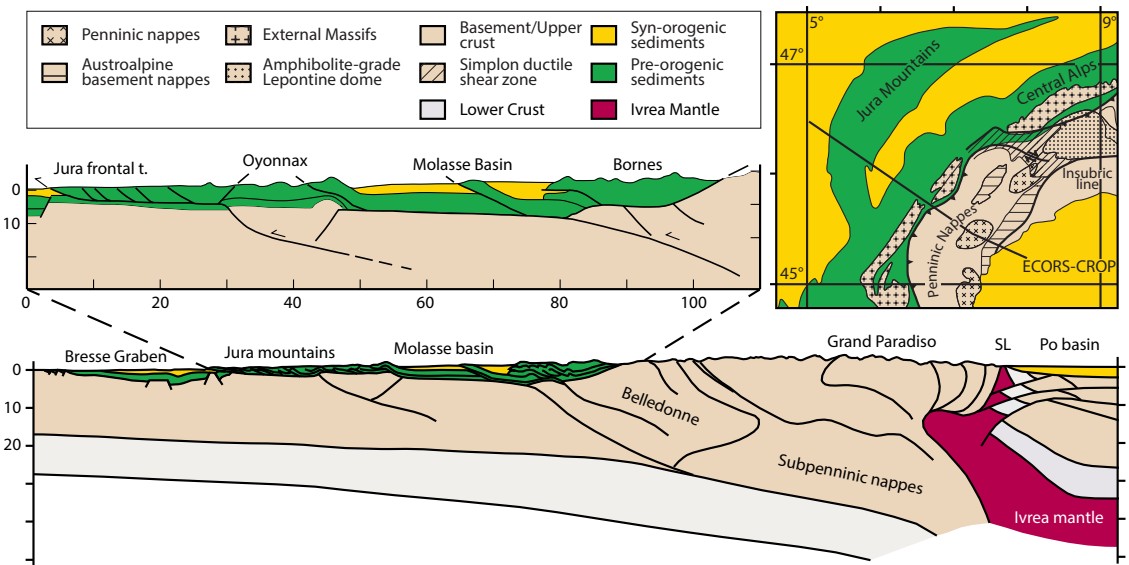

**Figure 1**





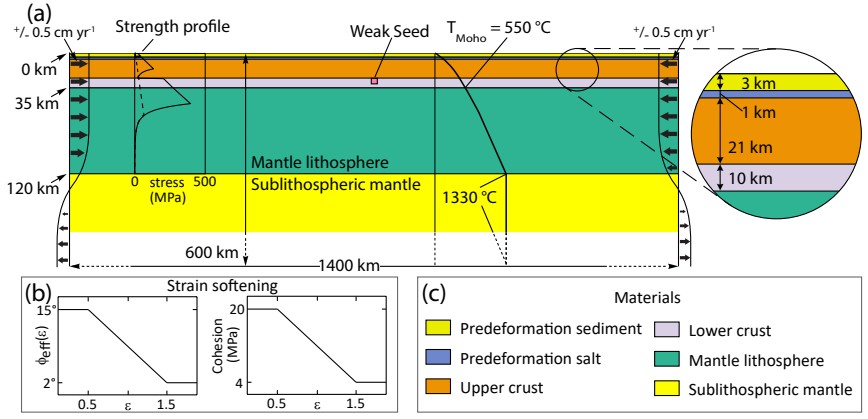

**Figure 2**







**Figure 3**





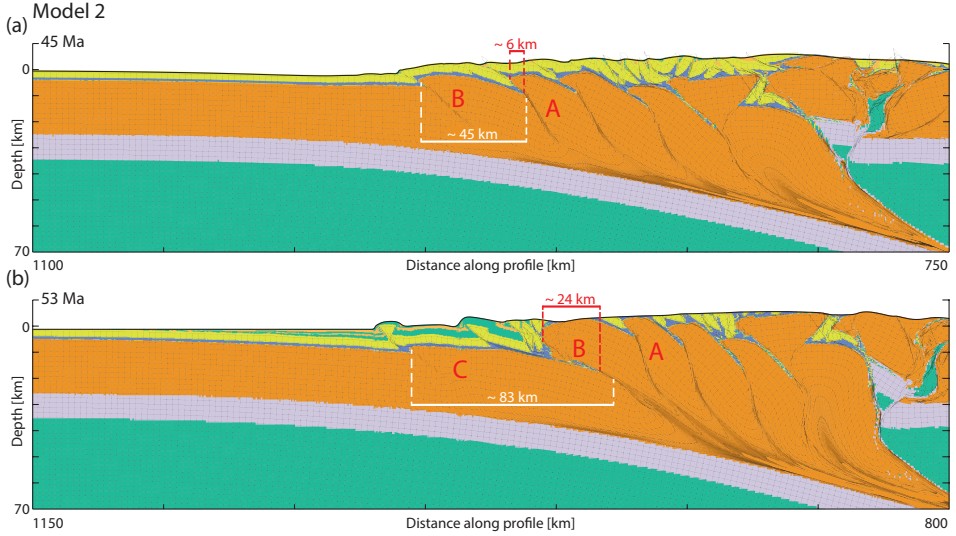

**Figure 4**



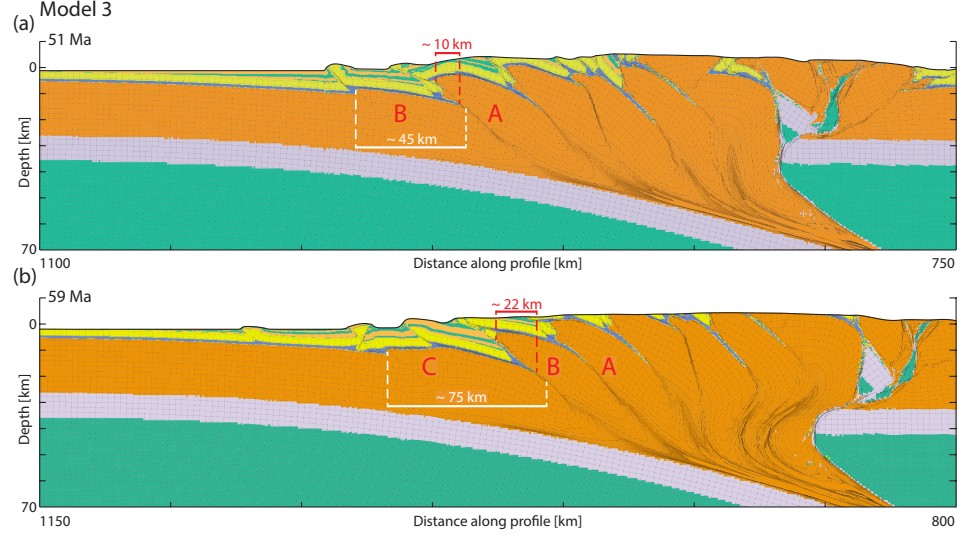

**Figure 5**



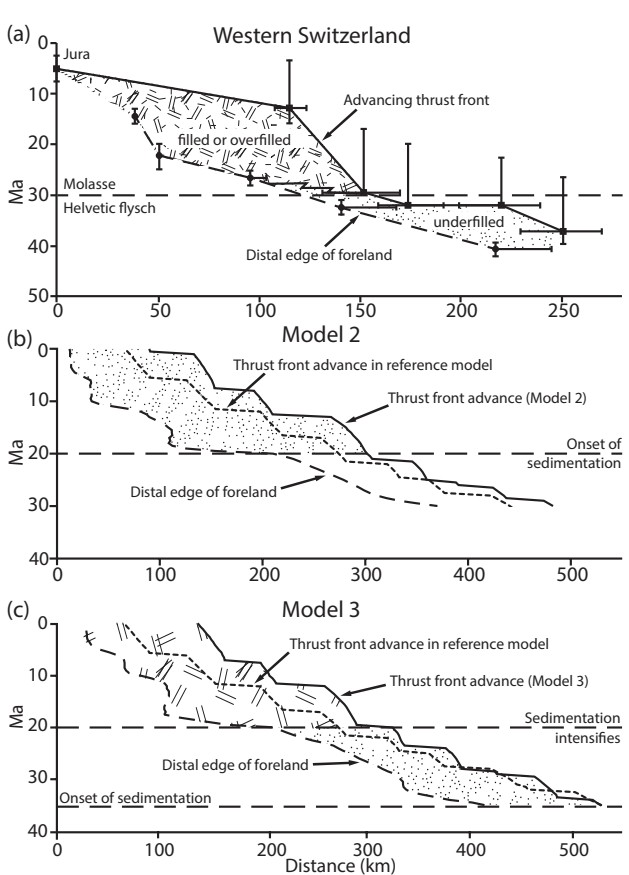

**Figure 6**





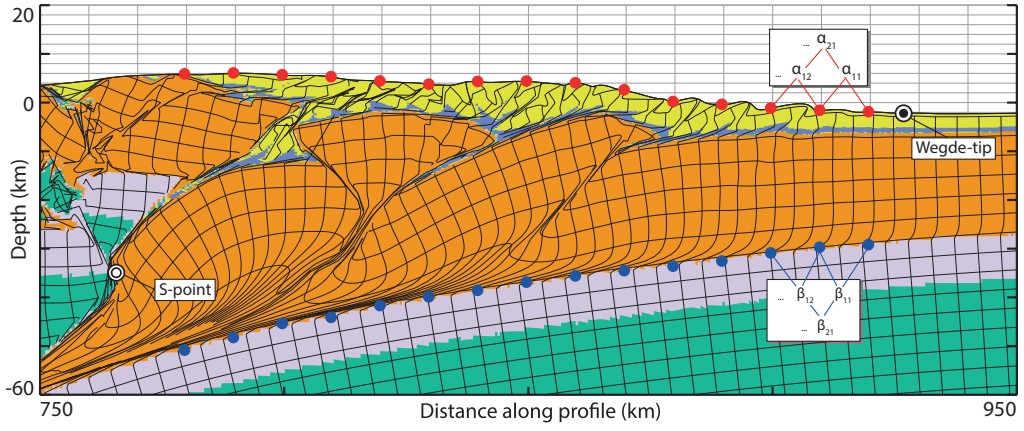

**Figure 7**





**Figure 8**





| Units | Salt | Upper Crust + Precollision Sediment | Lower Crust | Mantle Lithosphere | Sublithospheric Mantle |
|---|---|---|---|---|---|
| **Mechanical Parameters** | | | | | |
| Thickness (km) | 1 | 21 + 3 | 10 | 90 | 480 |
| Reference density (kg m$^{-3}$) | 2300 | 2800 | | 3360 | 3300 |
| Friction angle (deg) | - | 15° - 2° | | | |
| Cohesion (Pa) | - | $2.10^7 - 2.10^6$ | | | |
| Flow law | | Wet Quartz | | Dry Olivine | Wet Olivine |
| Reference | | Gleason and Tullis (1995) | | Karato and Wu (1993) | |
| scaling factor | 1 | 1 | 100 | 1 | 1 |
| A (Pa$^{-n}$ s$^{-1}$) | $8.574\ 10^{-28}$ | $8.574\ 10^{-28}$ | $8.574\ 10^{-28}$ | $2.4168\ 10^{-15}$ | $1.393\ 10^{-14}$ |
| Q (J mol$^{-1}$) | $222.815\ 10^3$ | $222.815\ 10^3$ | $222.815\ 10^3$ | $540.41\ 10^3$ | $429.83\ 10^3$ |
| n | 4 | 4 | 4 | 3.5 | 3 |
| V (m$^3$ mol$^{-1}$) | 0 | $3.1\ 10^{-6}$ | $3.1\ 10^{-6}$ | $25\ 10^{-6}$ | $15\ 10^{-6}$ |
| R (J mol$^{-1}$ °C$^{-1}$) | | 8.3144 | | | |
| **Thermal Parameters** | | | | | |
| Heat capacity (m$^2$ K$^{-1}$ s$^{-2}$) | | 803.57 | | 681.82 | |
| Thermal conductivity (W m$^{-1}$ K$^{-1}$) | | 2.25 | | | |
| Thermal exansion (K$^{-1}$) | | $3.1\ 10^{-5}$ | | 0 | |
| Heat productivity (μW m$^{-3}$) | | $0.8\ 10^{-6}$ | | 0 | |