# Peer review of "Control of increased sedimentation on orogenic fold-and-thrust belt structure – Insights into the evolution of the Western Alps"

_Solid Earth, 2018_

## Referee Comment (RC1) · Yang (Referee) · 25 Dec 2018

The manuscript, " Control of increased sedimentation on orogenic fold-and-thrust belt structure –Insights into the evolution of the Western Alps" by Erdos et al., presents an innovative numerical study investigating the evolution of orogenic fold-thrust belt structures with control of intensified surface process, i.e. sedimentation coupled with erosion. Their presented models show all detailed structures of thin-skinned and thick-skinned fold-thrust belts similar to that observed in natural examples, which is further used to investigate in high detail the development of basement thrusts and topography. I enjoyed reading this paper, which I think is a useful reference work for

the literature on fundamental thrust belt mechanics in various tectonic settings. The text is very well-written and easy to follow, I recommend only minor modifications.

Please also note the supplement to this comment:
https://www.solid-earth-discuss.net/se-2018-122/se-2018-122-RC1-supplement.pdf
* * *
[Figure]

**Supplement:**

Recommendation: Accept for publication with minor revision

The manuscript, " Control of increased sedimentation on orogenic fold-and-thrust belt structure –Insights into the evolution of the Western Alps" by Erdős et al., presents an innovative numerical study investigating the evolution of orogenic fold-thrust belt structures with control of intensified surface process, i.e. sedimentation coupled with erosion.  Their presented models show all detailed structures of thin-skinned and thick-skinned fold-thrust belts similar to that observed in natural examples, which is further used to investigate in high detail the development of basement thrusts and topography.

I enjoyed reading this paper, which I think is a useful reference work for the literature on fundamental thrust belt mechanics in various tectonic settings. The text is very well-written and easy to follow, I recommend only minor modifications.

Abstracts:

Page 2, Line 3-4: change "where a sudden increase in foreland sedimentation rate is well documented during the mid-Oligocene" to "where a sudden increase in foreland sedimentation rate during the mid-Oligocene is well documented"

Page 2, Line 5: "the basement thrust" should be "the frontal/outermost basement thrust". The study is focused on the frontal basement thrust development and propagation, so it is better to state this clear in the abstract. Otherwise it might be confusing with the early basement thrusts that may also remain minor active or be reactivated during the onset of increasing sedimentation.

Page 2, Line 10: "thrust-front propagation rate", here I assume the authors refer to "basement thrust front rate" not the "thin-skinned thrust front rate". So better to make the description precise and clear by adding basement in front of this phrase.

Introduction:

Page 3, Line 29: "model". The Critical Taper Theory is not just a model, but a widely accepted theory as the authors also described. So replace "model" by "theory".

Discussion;

Page 6, Line 23: "One of these basement thrusts", here needs further explanation on the particular basement thrust that then becomes the locus of subduction, such as its angle, displacement or position. Because the locus of subduction is of particular importance to the subsequent formation of new basement thrust in the subducting pro-wedge lithosphere.

Page 7: line 6-7: "a new thrust", "deformation front". Both need the "basement"  to be added to make the definition more specific. In the thin-skinned and thick skinned coupled

deformational system, the single description of a new thrust and deformation front can represent structural features at either thin-skinned or thick skinned belt.

Page 7, line 33 "new thrust", line 35 "thrust front", similar problem as raised above.

Page 8, line 6 "thrust front propagation", similar problem as raised above.

Page 9, line15-16: in this first sentence, the initiation behaviour of the new thin-skinned front thrusts between the reference Model 1 of no sedimentation and Model 2 and Model 3 of significant sedimentation is not comparable as the new thin-skinned thrust in the Figure 8a is missing. Add the new thin-skinned thrust in figure 8a and make the comparison.

Page 9: line 34: "This suggests that $\alpha \approx 10°$ can locally be seen as a critical value", the critical taper theory has provided robust equations to calculate the critical taper angle with given wedge material properties. Here the theory predicted value is needed and further discussion should be followed.

Figures captions:

Page 15: line 12: "The material coloring scheme is identical to that used in Figure 1". Figure 1 does not include colour scheme about the different modelling mechanical layers, so it should be Figure 2.

Page 15, line 26, same problem. Replace "Figure 1" by "Figure 2".

Page 15, line 34, please define "over-steeped", or just state "steepened".

Page 15: line 37: "Figure 1" to be replaced by "Figure 2".

Page 16: line 2: "Figure 3" to be replaced by "Figure 4".

Page 16: line 6: again either define "gradually steepening" or just use "steepening".

Figures:

Page 18 Figure 1, Jura Frontal t. needs to be clarified as "Jura Frontal thrust"

Page 20 Figure 3, figures a, d, g are missing horizontal distance. Figure b and c show an increasing distance from foreland to retro-side of orogen while the figure h shows the opposite distance trend. Please make the distance direction determined consistently across different model examples.

Page 23 Figure 6, after the onset of sedimentation, distal edge of foreland becomes more complex with some slightly retreat marked by concave line, i.e. 16-18 Ma and 100 km in figure b and 13-15 Ma and 100-105 km in figure c. These needs some explanation or discussion.

Page 25 Figure 8, each sub figure needs to project the predicted critical taper angle by Critical Taper Theory and make further discussion. Figure 8a needs the initiation of new thin-skinned thrusts within the pre-deformation sediments. The figure 8c shows two successive thick-skinned thrusts that form very closely in time, i.e. 41 My and 42 My respectively, which seems to contradict the statement that the frontal basement thrust becomes stationary after the onset of sedimentation. Please make some explanation on this. The unit at the horizontal axis should be consistent as that in the text "Myr".

Overall suggestions:

The study involves the structural behaviour of both foreland thin-skinned fold-thrust belt and basement fold-thrust belt, but there are many occasions where the description of thrust front, frontal thrust, deformation front are not confusing. All these need to be clarified, such as distal edge of foreland basin, basement deformation front, new thin-skinned thrust fault, new thick-skinned thrust fault, basement thrust fault, outermost/ frontal basement thrust fault.

A significant portion of texts associated with Figures 7 and 8 belongs to the result section, but they are completely described and discussed in the discussion section. I would suggest a separated result section on the taper angle of modelled fold-thrust belts and initiation of new think-skinned and thin skinned thrust faults.

The authors claims that the results presented in this study have broad implications for orogenic belts other than the Western Alps. It will be useful to add a schematic diagram to generalize the major conclusions of this study, which will help to gain a better and more direct application of this study to other tectonic settings.

Xiaodong Yang

25 December 2018

---

## Referee Comment (RC2) · Anonymous Referee #2 · 4 Jan 2019

This manuscript investigates the relationship between rapid synorogenic sediment filling and the development of foreland fold-and-thrust belts. It focuses in particular on the temporary slowing of thrust-front propagation, as observed in the North Alpine Foreland Basin. The results of this study are also compared to predictions of the critical taper theory. The manuscript is concise, well written and illustrated. However, I have two main questions that I would like to address:

1) The initial configuration of the numerical model is not specified. Could you please comment on why you have chosen this specific setup? And could you, if possible, give references to the Alps, i.e. extension/convergence rate, sedimentation rates, erosion

rates, crustal geometry, etc.. Changes in model parameters are likely to alter the model results. It would be therefore useful to know why you have chosen them in the first place.

2) Are the timescales observed in the models comparable to those observed in the Alpine Foreland Basin? Please elaborate.

Other comments:

Introduction Line 7-11: What effect does synorogenic sedimentation have on the development of thin- and thick-skinned foreland thrust sheets. Please elaborate.

Numerical method: Details on the crustal thickness, extension/compression velocities, thermal gradient and the position of the weak seed and are all shown in Fig. 2, but are not mentioned in the text. Is extension instantaneously followed by compression?

Model 1: I think it could be useful to give a brief definition of pro- and retro-side.

Model 2: What are the corresponding erosion rates?

Critical taper theory: This section appears a bit unconnected to the rest of the text. I suggest to rearrange parts of it., i.e. models described in this section could be moved to the result section.

---

## Author Comment (AC1) · 25 Jan 2019

Recommendation: Accept for publication with minor revision The manuscript, " Control of increased sedimentation on orogenic fold-and-thrust belt structure –Insights into the evolution of the Western Alps" by ErdÅŚs et al., presents an innovative numerical study investigating the evolution of orogenic fold-thrust belt structures with control of intensified surface process, i.e. sedimentation coupled with erosion. Their presented models show all detailed structures of thin-skinned and thick-skinned fold-thrust belts similar to that observed in natural examples, which is further used to investigate in high detail the development of basement thrusts and topography.

[Figure]

I enjoyed reading this paper, which I think is a useful reference work for the literature on fundamental thrust belt mechanics in various tectonic settings. The text is very well-written and easy to follow, I recommend only minor modifications.

Abstracts: Page 2, Line 3-4: change "where a sudden increase in foreland sedimentation rate is well documented during the mid-Oligocene" to "where a sudden increase in foreland sedimentation rate during the mid-Oligocene is well documented"

Response: We will implement the suggested rephrasing.

Page 2, Line 5: "the basement thrust" should be "the frontal/outermost basement thrust". The study is focused on the frontal basement thrust development and propagation, so it is better to state this clear in the abstract. Otherwise it might be confusing with the early basement thrusts that may also remain minor active or be reactivated during the onset of increasing sedimentation.

Response: We will implement the suggested clarification.

Page 2, Line 10: "thrust-front propagation rate", here I assume the authors refer to "basement thrust front rate" not the "thin-skinned thrust front rate". So better to make the description precise and clear by adding basement in front of this phrase.

Response: The reviewer is correct. We will implement the suggested clarification.

Introduction: Page 3, Line 29: "model". The Critical Taper Theory is not just a model, but a widely accepted theory as the authors also described. So replace "model" by "theory".

Response: We will implement the suggested rephrasing.

Discussion; Page 6, Line 23: "One of these basement thrusts", here needs further explanation on the particular basement thrust that then becomes the locus of subduction, such as its angle, displacement or position. Because the locus of subduction is of particular importance to the subsequent formation of new basement thrust in the

subducting pro-wedge lithosphere.

Response: We will clarify the text based on the reviewer's suggestion.

Page 7: line 6-7: "a new thrust", "deformation front". Both need the "basement" to be added to make the definition more specific. In the thin-skinned and thick skinned coupled deformational system, the single description of a new thrust and deformation front can represent structural features at either thin-skinned or thick skinned belt.

Response: We will implement the suggested rephrasing.

Page 7, line 33 "new thrust", line 35 "thrust front", similar problem as raised above.

Response: We will implement the suggested rephrasing.

Page 8, line 6 "thrust front propagation", similar problem as raised above.

Response: We will implement the suggested rephrasing.

Page 9, line15-16: in this first sentence, the initiation behaviour of the new thin-skinned front thrusts between the reference Model 1 of no sedimentation and Model 2 and Model 3 of significant sedimentation is not comparable as the new thin-skinned thrust in the Figure 8a is missing. Add the new thin-skinned thrust in figure 8a and make the comparison.

Response: In case there is only the pre-orogenic sediments available for thin-skinned deformation, a new thin-skinned thrust form every 0.5 Myr or so. Marking this on figure 8a would make the figure indecipherable.

Page 9: line 34: "This suggests that $\alpha \approx 10°$ can locally be seen as a critical value", the critical taper theory has provided robust equations to calculate the critical taper angle with given wedge material properties. Here the theory predicted value is needed and further discussion should be followed. Response: We cannot strictly apply critical taper theory because (1) the material parameters (in particular $\mu$) are not constant in space and time due to strain weakening; and (2) the model includes ductile (viscous)

behavior. Additionally, we cannot derive a value from our models for a number of parameters of the critical taper equation (e.g. the basal pore fluid to lithostatic pressure ratio, compressive wedge strength), hence even if we neglected the above mentioned complexities, we cannot solve the equation.

Figures captions: Page 15: line 12: "The material coloring scheme is identical to that used in Figure 1". Figure 1 does not include colour scheme about the different modelling mechanical layers, so it should be Figure 2.

Response: The reviewer is correct. The caption should refer to figure 2. This will be modified in the text.

Page 15, line 26, same problem. Replace "Figure 1" by "Figure 2".

Response: The reviewer is correct. The caption should refer to figure 2.

Page 15, line 34, please define "over-steeped", or just state "steepened".

Response: We will implement the suggested rephrasing.

Page 15: line 37: "Figure 1" to be replaced by "Figure 2".

Response: The reviewer is correct. The caption should refer to figure 2.

Page 16: line 2: "Figure 3" to be replaced by "Figure 4".

Response: The reviewer is correct. The caption should refer to figure 4.

Page 16: line 6: again either define "gradually steepening" or just use "steepening".

Response: We will implement the suggested rephrasing.

Page 18 Figure 1, Jura Frontal t. needs to be clarified as "Jura Frontal thrust"

Response: We will implement the suggested clarification.

Page 20 Figure 3, figures a, d, g are missing horizontal distance. Figure b and c show an increasing distance from foreland to retro-side of orogen while the figure h shows the

opposite distance trend. Please make the distance direction determined consistently across different model examples.

Response: The horizontal scale for the large-scale subfigures (a, d, g, i) is the same, and stated only once on the bottom of subfigure i. Generally, we use a left-to-right horizontal scale for all models but since the initial model setup is perfectly symmetrical, the direction of subduction is random. For ease of comparison we flipped models 2 and 3 to show them in the same orientation that is conventional for the alpine cross-sections. Based on the reviewer's comment, we clarify this in the caption.

Page 23 Figure 6, after the onset of sedimentation, distal edge of foreland becomes more complex with some slightly retreat marked by concave line, i.e. 16-18 Ma and 100 km in figure b and 13-15 Ma and 100-105 km in figure c. These needs some explanation or discussion. Response: The reviewer is correct. This feature needs some explanation. We clarify this in the text. Page 25 Figure 8, each sub figure needs to project the predicted critical taper angle by Critical Taper Theory and make further discussion. Figure 8a needs the initiation of new thin-skinned thrusts within the pre-deformation sediments. The figure 8c shows two successive thick-skinned thrusts that form very closely in time, i.e. 41 My and 42 My respectively, which seems to contradict the statement that the frontal basement thrust becomes stationary after the onset of sedimentation. Please make some explanation on this. The unit at the horizontal axis should be consistent as that in the text "Myr".

Response: For the first comment on the Critical Taper Theory bit, we refer to our earlier response on the issue. For the second part of the comment: In case there is only the pre-orogenic sediments available for thin-skinned deformation, a new thin-skinned thrust form every 0.5 Myr or so. Marking this on figure 8a would make the figure indecipherable. Regarding figure 8c: the two basement thrusts in question form 6 and 7 Myr after the onset of sedimentation so their existence does not contradict our statement about the stalling of basement thrust propagation. The second one of the two is a bit special as it seems to be localized by the rather long-lived thin-skinned

frontal thrust that becomes active after the onset of sedimentation. The reviewer is correct about the inconsistency of the units. We address this on the figure.

Overall suggestions: The study involves the structural behaviour of both foreland thin-skinned fold-thrust belt and basement fold-thrust belt, but there are many occasions where the description of thrust front, frontal thrust, deformation front are not confusing. All these need to be clarified, such as distal edge of foreland basin, basement deformation front, new thin-skinned thrust fault, new thick-skinned thrust fault, basement thrust fault, outermost/ frontal basement thrust fault.

Response: Based on the reviewer's suggestion, we will clarify the text.

A significant portion of texts associated with Figures 7 and 8 belongs to the result section, but they are completely described and discussed in the discussion section. I would suggest a separated result section on the taper angle of modelled fold-thrust belts and initiation of new think-skinned and thin skinned thrust faults.

Response: since both reviewers made this point, we will restructure the text so that the critical taper theory part will become its own section between the results and the discussion sections as we think it would be difficult and possibly counterproductive to separate the results from the interpretation.

The authors claims that the results presented in this study have broad implications for orogenic belts other than the Western Alps. It will be useful to add a schematic diagram to generalize the major conclusions of this study, which will help to gain a better and more direct application of this study to other tectonic settings. Response: In accordance with the reviewer's suggestion, we will create an additional figure generalize the major conclusions.
* * *

---

## Author Comment (AC2) · 25 Jan 2019

This manuscript investigates the relationship between rapid synorogenic sediment filling and the development of foreland fold-and-thrust belts. It focuses in particular on the temporary slowing of thrust-front propagation, as observed in the North Alpine Foreland Basin. The results of this study are also compared to predictions of the critical taper theory. The manuscript is concise, well written and illustrated. However, I have two main questions that I would like to address:

1) The initial configuration of the numerical model is not specified. Could you please comment on why you have chosen this specific setup? And could you, if possible, give

references to the Alps, i.e. extension/convergence rate, sedimentation rates, erosion rates, crustal geometry, etc.. Changes in model parameters are likely to alter the model results. It would be therefore useful to know why you have chosen them in the first place.

Response: The initial parameters (crustal setup, convergence velocity) have been chosen to match the circumstances likely applicable for the Pyrenean-Alpine orogenic systems. The major differences are that there is no ocean and no lag between break-up and onset of inversion, but these differences are due to computational limitations. The sedimentation/erosion algorithms are rather simplistic and have been parametrized to represent moderate rates for both processes. Based on the reviewer's comment we will clarify this in the text.

2) Are the timescales observed in the models comparable to those observed in the Alpine Foreland Basin? Please elaborate.

Response: The shortening rates, timing of orogenesis and of transition from an under- to an over-filled basin are based on observations in the Alps and the timescales of thrust/basin evolution are comparable. The jump in thrust-front position is in the order of a 100 km in both cases, but the stagnation in the Alps lasted about twice as long as what we observe in the models. Based on the reviewer's comment we will clarify this in the text.

Other comments: Introduction Line 7-11: What effect does synorogenic sedimentation have on the development of thin- and thick-skinned foreland thrust sheets. Please elaborate.

Response: we will expand the topic based on the reviewer's comment.

Numerical method: Details on the crustal thickness, extension/compression velocities, thermal gradient and the position of the weak seed and are all shown in Fig. 2, but are not mentioned in the text. Is extension instantaneously followed by compression?

Response: We will expand the text based on the reviewer's comment but some of the above-mentioned details are covered in the supplementary materials. The extension is instantaneously followed by compression (i.e. there is no thermal relaxation phase). This limitation is covered in the discussion.

Model 1: I think it could be useful to give a brief definition of pro- and retro-side.

Response: We will expand the text based on the reviewer's comment.

Model 2: What are the corresponding erosion rates?

Response: This information is covered in Supplement 1. The elevation dependent erosion algorithm is scaled so that a 2-km high topography erodes by 1 km in 1 Myr.

Critical taper theory: This section appears a bit unconnected to the rest of the text. I suggest to rearrange parts of it., i.e. models described in this section could be moved to the result section.

Response: since both reviewers made this point, we will restructure the text so that the critical taper theory part will become its own section between the results and the discussion sections as we think it would be difficult and possibly counterproductive to separate the results from the interpretation.